# Autophagy in the Lifetime of Plants: From Seed to Seed

**DOI:** 10.3390/ijms231911410

**Published:** 2022-09-27

**Authors:** Song Wang, Weiming Hu, Fen Liu

**Affiliations:** Lushan Botanical Garden, Chinese Academy of Sciences, Jiujiang 332900, China

**Keywords:** autophagy, abiotic stress, biotic stress, vegetative growth, reproductive growth

## Abstract

Autophagy is a highly conserved self-degradation mechanism in eukaryotes. Excess or harmful intracellular content can be encapsulated by double-membrane autophagic vacuoles and transferred to vacuoles for degradation in plants. Current research shows three types of autophagy in plants, with macroautophagy being the most important autophagic degradation pathway. Until now, more than 40 autophagy-related (ATG) proteins have been identified in plants that are involved in macroautophagy, and these proteins play an important role in plant growth regulation and stress responses. In this review, we mainly introduce the research progress of autophagy in plant vegetative growth (roots and leaves), reproductive growth (pollen), and resistance to biotic (viruses, bacteria, and fungi) and abiotic stresses (nutrients, drought, salt, cold, and heat stress), and we discuss the application direction of plant autophagy in the future.

## 1. Introduction

Autophagy, also known as self-eating, is an evolutionarily conserved process that occurs in eukaryotic cells and involves the degradation of organelles, protein complexes, and macromolecules [1]. Generally, the degraded material is sequestered into autophagic vesicles that are transported to the vacuole for breakdown. Under normal conditions, autophagy is a housekeeping process that degrades unwanted cytoplasmic content and maintains cellular homeostasis [2]. Under stress conditions (starvation, oxidative and abiotic stress, and pathogen infection), autophagy proteins are up-regulated and help in recycling damaged or non-essential cellular material [3].

Until now, three different types of autophagy in plants have been discovered, including microautophagy, macroautophagy, and mega-autophagy [4]. Microautophagy is the direct packaging of cargo into the vacuole for degradation through the invagination or protrusion of the vacuolar membrane [5]. Although the concept of microautophagy has been present for many years, little is known about the mechanism by which it occurs. In plants, microautophagy is found to play an important role in anthocyanin aggregates [6]. In addition, microautophagy is also involved in the degradation progress of damaged chloroplast, namely, chlorophagy [7]. The most well-studied type of autophagy in plants is macroautophagy, in which autophagosomes form and then fuse with vacuoles to degrade cargoes [8]. Until now, more than 40 ATG proteins have been found to be involved in the biological process of macroautophagy [1]. There is also a more direct type of autophagy, namely, mega-autophagy. Here, the tonoplast membrane ruptures to release the vacuolar hydrolases directly into the cytoplasm, where it degrades cytoplasmic materials [9,10]. Mega-autophagy often occurs during programmed cell death (PCD) including plant development or in response to pathogens [1].

In plants, autophagy is a feature in diverse biological processes such as development, nutrient recycling, and biotic and abiotic stresses [2] (Figure 1). It can be seen that autophagy plays a pivotal role in the life of plants. In this review, we summarize the research progress regarding how autophagy affects plant lifespan, especially concerning growth and development, resistance to abiotic stress, and interaction with microorganisms.

## 2. Autophagy in Vegetative Growth

### 2.1. Seed Development

For flowering plants, the cycle of life begins with a seed. Generally, *atg* mutations in plants produce fewer seeds compared to WT plants, suggesting that autophagy may function during plant seed development [11]. In Arabidopsis, several *atg* mutations show decreased seed production [2], and some *ATG* genes are up-regulated during seed maturation [12]. Similar results are shown in maize, such as *ATG1a*, *Atg18e*, *Atg18e*, *Atg18f*, and *Atg18h*, which are expressed in endosperm instead of other tissues [13]. However, the role of autophagy in seed development has not been explained at the mechanism level. Some studies show that autophagy may contribute to the transport of seed storage proteins [14,15]. Both total protein and 12S globulins are accumulated in *atg5* seeds, indicating that autophagy affects the seed protein content [15]. In addition to seed protein accumulation, autophagy also contributes to seed germination. In Arabidopsis, the overexpression of Atg8-interacting proteins (ATI1 and ATI2) can stimulate seed germination under ABA conditions [16].

Some research shows that autophagy is also involved in plant oil production. In Brassicaceae, oil accumulates in embryonic tissues and endosperm during seed development and seed germination [17]. In addition, lipid droplets (LDs) need to be degraded under specific metabolic or physiological conditions, and this process occurs mainly via autophagy [18]. In Arabidopsis, the overexpression of *ATG5* or *ATG7* increases both the seed yield and the fatty acid (FA) content [11]. This process may not directly affect oil metabolism in seeds, but does affect the re-mobilization of resources from leaves [19].

### 2.2. Root Development

Structurally, plant roots are divided into three zones: the meristematic zone, elongation zone, and maturation zone [20]. A cross-section of roots includes three levels: dermal, cortex (ground tissue), and vascular tissues [21]. During root development, autophagy plays an important role in its establishment and functional differentiation.

*ATG8* genes are mainly located in the root caps and maturation zone, which correspond to relevant protein degradation [22]. The role of autophagy in the root tips may be related to programmed cell death (PCD), but there is not enough evidence to prove this speculation [21].

Some evidence suggests that autophagy is involved in the formation of cortical tissue. Cortical parenchyma cells contain a large vacuole. In Arabidopsis roots, autophagy is required for ground tissue differentiation, in order to degrade cytoplasmic material and for vacuole formation from the meristem to the elongation zone [23]. In ground tissue cells, partial cytoplasmic accumulation is observed in central vacuoles, suggesting that autophagy occurs in these regions [23]. In Arabidopsis roots, the ATG8f protein localizes to autophagy-like structures in the central vacuole, suggesting that autophagy also determines vacuolar generation in cortical parenchymal cells [22].

Autophagy also plays a significant role in the differentiation of vascular tissue (xylem and phloem). In roots, xylem formation occurs primarily via nitric oxide (NO) signaling, through the synthesis of secondary cell walls and degradation of protoplast [24]. Autophagy-related processes appear to play a role in central vacuole formation and the degradation of cytoplasmic material at the onset of xylem partialization [21]. Several *ATG* genes (*ATG8C*, *ATG8D*, *ATG11*, and *ATG18*) are certified in the differentiation of the xylem [25]. Compared to the xylogenesis, there is a lack of discussion in the literature reporting on the mechanism of autophagy during phloemogenesis. There is a report indicating that ATG8 protein localization can be observed in differentiating the primary phloem [25].

In addition to root development, autophagy also plays a vital role in root senescence [26]. The up-regulation of *ATG* genes (*ATG8C*, *ATG8D*, and *ATG8G*) is characterized by the senescence of absorptive roots [26]. During the first stage of senescence, autophagy counteracts transient cell death and maintains cellular homeostasis [21]. Additionally, autophagy is also involved in the remobilization process, which is a key step in the senescence process [21].

### 2.3. Leaf Senescence

Leaf senescence is a late stage of plant vegetative growth. In Arabidopsis, premature leaf senescence is one of the common phenotypes in autophagy mutants. Most *ATG* genes are up-regulated under leaf senescence, while other autophagy genes are mainly expressed in leaf development. In barley, the transcript levels of both *ATG7* and *ATG18f* are up-regulated during leaf senescence [27]. In rice, compared to the wild-type, the *atg7* mutant not only reduces the plant height, root length, tiller number, and leaf area, but also has obvious premature leaf senescence [28]. In Arabidopsis, *SAG12* (senescence-associated gene 12) is a marker gene for the onset of senescence, which is abundantly induced in *atg2* and *atg5* mutants [29]. Premature leaf senescence of these mutants can be alleviated by blocking SA (salicylic acid) biosynthesis or signal transduction. For example, the overexpression of SA hydroxylase *NahG* (salicylate hydroxylase) can inhibit the premature senescence phenotype of *atg2* and *atg5*, and the administration of the SA analog BTH (benzothiadiazole) restores the normal phenotype of these mutants [29]. The regulation of autophagy in premature plant senescence may be attributed to the effect on the redistribution of plant nutrients, especially nitrogen. For example, rice *atg7–1* mutant leaves prematurely senesce, and the nitrogen content of senescent leaves is higher than that of wild-type leaves, which reduces the nitrogen reuse efficiency [28]. In apples, the overexpression of ATG18a greatly improves resistance to low nitrogen stress and up-regulates the expression of nitrogen uptake and assimilation-related genes *NIA2*, *NRT2.1*, *NRT2.4*, and *NRT2.5* [30]. Some new evidence suggests that *SAG12* regulates plant senescence through involvement in protein degradation and N remobilization [31,32]. In future studies, it will be interesting to determine whether autophagy cooperates with senescence-associated proteases to cycle cellular components.

## 3. Autophagy in Reproductive Growth

Autophagy is not only involved in regulating the vegetative growth, but also regulates the reproductive growth in plants. Members of the PI3K complex (*atg6*, *vps15*, and *vps34*) are reported to regulate Arabidopsis pollen maturation, and none of them can produce mature pollen after mutation [33,34,35]. In the rice *atg7* mutant, lipid and starch components in pollen grains cannot be accumulated normally during flowering, resulting in reduced pollen viability and sporophytic male sterility [36]. Autophagy also plays a key role in tobacco pollen germination. Autophagy flux is significantly increased in the early stage of pollen germination to degrade the cytoplasm in the germinal pores [37]. Cytoplasmic degradation of germinal pores during pollen germination is also inhibited after the silencing of *ATG2*, *ATG5*, and *ATG7* in tobacco [38]. New research shows that autophagy is also involved in pollen tube elongation. In this process, a core protein SH3-domain-containing protein 2 (SH3P2) colocalizes with ATG proteins and participates in regulating mitophagosomes [38]. Down-regulation of SH3P2 expression significantly impairs pollen germination and pollen tube growth [38].

## 4. Autophagy in Abiotic Stress

Plants are exposed to various abiotic stresses during growth, such as salt, heat, cold, drought, and nutrition stress. Autophagy, a process that maintains cellular homeostasis, plays an important role in the defense of abiotic stresses.

### 4.1. Autophagy under Nutrient Starvation

Nutrient starvation triggers a strong induction of autophagy, and *ATG* mutants exhibit premature senescence upon carbon/nitrogen starvation. Using sucrose starvation in suspension-cultured cells shows that 30% to 50% of the total protein is degraded over a two day period [39], and the decrease in total proteins stems from non-selective degradation, rather than degradation of specific proteins [40]. Fusion between autophagosomes and central vacuoles is observed in cells treated with E-64c (cysteine protease inhibitors). Thus, during classical autophagy, the partially degraded cytoplasm in the autophagosome is likely to be released into the central vacuole for further degradation [40]. Using bafilomycin A_1_ and concanamycin A, which inhibits the activity of vacuolar hydrolase, it is observed that even non-degradable autophagosomes are still expelled into the central vacuole [41]. Nutrient stress also affects target of rapamycin (TOR) signaling and ultimately activates autophagy production. For example, autophagy induced by nutrient stress is inhibited in TOR-overexpressing plants, while the application of the TOR inhibitor AZD8055 in wild-type plants and *raptor1a/b* mutants leads to the production of structural autophagy [42]. Studies found that, although the ATG1 complex is involved in autophagy induced by nitrogen starvation and short-term carbon starvation, there is an ATG1-independent autophagy initiation pathway under long-term carbon starvation in Arabidopsis, in which the SnRK1 catalytic subunit KIN10 can directly phosphorylate the ATG6 to initiate autophagy [43].

In addition to N and C starvation, studies show that autophagy also plays an important role in other nutrient stresses. A previous study shows that autophagy can balance zinc (Zn) and iron (Fe) uptake in plants [44]. Previous studies also show that autophagy can increase zinc bioavailability in plants when zinc levels in the environment are low [45]. *atg* mutants exhibit more severe chlorosis compared to the wild-type under zinc-deficient conditions [45]. Zn deficiency not only induces autophagy to degrade various targets, but also observes that the amount of mobile Zn^2+^ in *atg* mutants is much lower than in the WT under Zn starvation conditions [45]. Interestingly, autophagy is also induced when zinc is in excess, and provides mobile iron ions from a non-mobile bound form to balance zinc–iron homeostasis in plants [46]. In this Zn–Fe balance process, bZIP19 and bZIP23 may be the switches to initiate/inhibit autophagy in plants under the condition of Zn deficiency/excess, and BRUTUS (BTS) may be involved in the initiation of autophagy under Fe deficiency [44]. A new study found that autophagy is also involved in the phosphate (Pi) response. However, the effect of Pi starvation on the degradation process of autophagy-regulated proteins is slight, and only a few proteins can be degraded, which can be used as characteristic target proteins under phosphorus starvation conditions for phosphorus deficiency-related studies [47].

A new study shows that autophagy is also involved in the regulation of sulfur starvation in plants [48]. Sulfur (S) remigration from the rosette to the seed is impaired in *atg5* mutants compared to the wild-type [48]. These studies demonstrate that autophagy plays an indispensable role in maintaining cell homeostasis in plants under nutrient starvation.

### 4.2. Drought Stress

Drought stress increases the expression of many *ATG* genes in crops, such as *ATG2* in peppers [49], *ATG8a* in millet [50], *ATG6* in barley [51], and *ATG3* and *ATG18a* in apples [28,52]. In tomatoes overexpressing *HsfA1a*, the silencing of *ATG10* and *ATG18f* reduces *HsfA1a*-induced drought tolerance and autophagosome formation [53]. Conversely, the overexpression of *MdATG18a* in tomatoes degrades protein aggregation, limits oxidative damage, and ultimately improves drought tolerance [54]. Under drought stress, MtCAS31 promotes degradation of the MtPIP2;7 protein by autophagy, a negative regulator of drought, leading to a decrease in root hydraulic conductivity, thereby reducing water loss and improving drought tolerance [55]. In Arabidopsis, a plant-specific gene COST1, which is a negative regulator of drought, negatively regulates drought resistance by influencing the autophagy pathway [56]. COST1 co-localizes with ATG8e and the autophagy linker NBR1 on autophagosomes, suggesting a critical role in the direct regulation of autophagy [56]. A previous study found that mitochondrial alternative oxidase (AOX) may regulate autophagy through mitochondrial ROS during drought stress in tomatoes [57].

### 4.3. Heat and Cold Stress

Plants accumulate a large amount of oxidized and insoluble proteins at unsuitable temperatures. In this case, plants can eliminate these toxic proteins by inducing autophagy to improve plant resistance. *ATG* gene expression is up-regulated in various plants, and more autophagosomes are accumulated under heat stress [49,58,59]. On the contrary, silencing *ATG5* or *ATG7* in Arabidopsis and tomatoes under heat stress leads to heat sensitivity [58,59,60]. Plants degrade related proteins through NBR1-mediated selective autophagy under heat stress. Under heat stress, the expression of *NBR1* is up-regulated and more puncta accumulate in the cytoplasm compared to the wild-type [61]. Furthermore, more NBR1 puncta accumulate in WT plants during the heat stress recovery stage, and the accumulation of the NBR1 protein is significantly higher in *atg* mutants than in WT plants [61,62]. Furthermore, the NBR1-mediated selective autophagy pathway degrades HSP90.1 and ROF1 to reduce plant resistance to heat stress memory [63].

Compared to heat stress, there are few studies on the regulation of cold stress by autophagy in plants. In rice, *OsATG6b* is down-regulated under cold stress, while *OsATG6c* expression is up-regulated [51]. In barley, the expression of *HvATG6* is up-regulated under low temperatures [64]. This may suggest that ATG6 plays an important role in response to low plant temperature. In addition, NBR1-mediated selective autophagy also appears to be involved in plant responses to cold stress. In tomatoes, BRs (Brassinosteroids) and the positive regulator BZR1 induce autophagy and accumulation of the selective autophagy receptor NBR1 under cold stress [65].

### 4.4. Salt Stress

High concentrations of NaCl lead to a reduced photosynthetic rate, as well as excessive energy consumption and accumulation of excess reactive oxygen species (ROS) [66]. As an important regulator of cellular homeostasis, autophagy is also involved in the pathway of plant salt tolerance. Several autophagy genes are up-regulated under salt treatment in wheat seedlings [67]. Silencing metacaspase TaMCA-Id can reduce the tolerance of wheat seedlings to NaCl by promoting ROS production, which further participates in the regulation of autophagy and PCD triggered by NaCl treatment [68]. Within 3 h of salt treatment, accumulation of oxidized proteins in *atg2* and *atg7* is higher than that in the WT, and the mutants are highly sensitive to salt stress [69]. The Arabidopsis PI3K complex positively regulates salt tolerance by promoting the internalization of PIP2;1 from the plasma membrane into the vacuole under salt stress, thereby reducing root water permeability [70]. In addition, the overexpression of *MdATG10* leads to increased autophagy activity in roots and enhances salt tolerance in apples [71]. Another study demonstrates spermidine (Spd), a kind of polyamine, activation of *ATG* gene expression and autophagosome formation under salt stress in cucumbers [72]. All of the above results indicate the role of autophagy in plants under salt stress.

## 5. Autophagy in Biotic Stress

Besides abiotic stresses, biotic stresses can influence autophagy. In plants, depending on the lifestyle of the pathogen, infecting the plant and activating autophagy is shown to lead to different outcomes [73]. The plant immune system is a complex mechanism, the most well-known of which is the hypersensitive response (HR)-related programmed cell death (PCD) [74]. It is reported that autophagy is involved in plant immunity by negatively regulating PCD [75].

### 5.1. Autophagy in Plant Viral Infection

Autophagy is a double-edged sword regarding defense against plant viruses. Plant autophagy is both antiviral and can be manipulated or inhibited by plant viruses to facilitate viral infection [76]. Meanwhile, autophagy can maintain a dynamic balance between viral infection and host survival during pathogen infection [76].

Some studies show that autophagy can defend DNA viruses from infecting plants [77]. The virulence factor βC1 from CLCuMuV is degraded by autophagy through interaction with ATG8. Thus, viral infection is enhanced in *ATG7* and *ATG5* mutants [78]. βC1 acts as the first plant viral activator of autophagy to activate autophagy by disrupting the interaction between *ATG3* and glyceraldehyde-3-phosphate dehydrogenase, a negative regulator of autophagy [78]. In tomatoes, leaf curl Yunnan virus (TLCYnV) is degraded by autophagy through interaction with ATG8h [79]. In addition, infection by the double-stranded DNA pararetrovirus cauliflower mosaic virus (CaMV) also mediates its degradation through NBR1-regulated selective autophagy [80].

In addition to plant DNA viruses, autophagy has antiviral effects during infection by positive-strand RNA viruses. For example, the overexpression of *Beclin1/ATG6* inhibits TuMV viral RNA accumulation, while the knockout of *Beclin1* or *ATG8a* promotes its infection [81]. However, the molecular mechanism of autophagy inhibiting TuMV infection is still to be investigated further. A new study indicates that TuMV activates NBR1–ATG8f autophagy to target virus replication in the tonoplast for viral replication and accumulation [82].

Furthermore, autophagy also plays an antiviral role in negative-strand RNA virus infection. The viral suppressors of the RNAi (VSR) protein P3 from rice stripe virus (RSV) can interact with NbPI3P and can be degraded by autophagy, thereby inhibiting RSV infection [83]. In this process, eukaryotic translation initiation factor 4A (eIF4A) acts as a repressor by interacting with ATG5 to leave the ATG5–ATG12 interaction to inhibit autophagy [84]. From these current studies, autophagy can inhibit viral movement, not replication.

The above studies show that autophagy can suppress viral infection, but autophagy can also promote viral infection. For example, the overexpression of *ATG5* and *ATG8f* promotes the virus bamboo mosaic virus (BaMV) infection [85]. This may be because ATG8f-rich virus-induced vesicles can provide sites for viral RNA replication or evade host-silencing mechanisms [85].

Other studies found that autophagy can be manipulated by certain viruses. The γb protein from the barley stripe mosaic virus (BSMV) directly binds to ATG7 and blocks the interaction of ATG7 with ATG8 to inhibit autophagosome formation [86]. In addition, the turnip crinkle virus (TCV) uses the viral silencing repressor protein P38 to inhibit antiviral autophagy, possibly by directly sequestering the ATG8 protein [87].

### 5.2. Autophagy and Fungi

Fungal infection is one of the main hazards of plants and poses a major threat to food security. Some fungi, such as the rice blast pathogen, infect leaves by forming a dome-shaped cellular structure called an appressorium to break through the cuticle and cell wall of the host [88]. The formation of appressorium requires the accumulation of glycerol to establish cell expansion, and the energy required for the accumulation of glycerol needs to be transported from adjacent conidial cells that undergo autophagy-related cell death [89]. Therefore, autophagy mutants cannot produce appressorium to penetrate the host. For example, the necrotizing plant pathogen *Botrytis cinerea* cannot form an appressorium to infect plant cells after mutating the *atg1* gene [90]. In another study, a knockout mutant of a small Rab GTPases called MoYPT7 is shown to be impaired in autophagy and appressorium development in *Magnaporthe oryzae* [91]. In *Magnaporthe oryzae*, ATG1, ATG2, ATG3, ATG17, and ATG18 are shown to increase phosphorylation during appressorium formation, suggesting that post-translational modifications of ATG are involved in infecting the host [92]. Some reports found that host autophagy is also important for beneficial fungi. For example, the mycorrhizal fungus *Glomus intraradices* up-regulates the expression of *ATG8f* and *ATG4a* in the cortical cells and arbuscular cells of mycorrhizal colonized roots [93].

Autophagy can improve plant resistance to necrotrophic pathogens by limiting the hypersensitive response (HR) of host cells. Compared to the wild-type, multiple autophagy mutants in Arabidopsis show reduced resistance to *Botrytis cinerea*, specifically, severe leaf yellowing, larger lesion area, and more dead cells [94]. *AtATG18a* can positively regulate Arabidopsis necrotrophic pathogen immunity by interacting with *WRKY33*; at the same time, autophagy can induce the expression of PDF1.2 in jasmonic acid (JA) defense signals, which synergistically increases plant resistance to *Botrytis cinerea* [94]. Studies show that autophagy also has a positive regulatory role in response to biotrophic pathogens [95]. Silencing of *TaATG8j* results in the suppression of the wheat HR response after infection with *Puccinia striiformis* f. sp. *tritici*, resulting in reduced resistance [96]. Interestingly, knockout of *ATG6* in wheat increases the resistance of sensitive lines to *Blumeria graminis* f. sp. *tritici*, while knockout of *ATG6* in lines carrying the resistance gene *Pm21* reduces resistance, indicating that *ATG6* plays a complex role in wheat powdery mildew resistance [97].

### 5.3. Autophagy and Bacteria

Bacteria have evolved in many ways to manipulate host cells for successful infection. Numerous studies show that intracellular bacteria can manipulate autophagy as a pro-survival strategy [98].

*Pseudomonas syringae* is a Gram-negative bacterium with strong aerobic and saprotrophic properties, and the incidence of plant diseases caused by *Pseudomonas syringae* ranks first in the top 10 bacterial plant diseases [99]. Some studies show that autophagy has reverse resistance to *Pseudomonas syringae*. Arabidopsis *atg5*, *atg18a* mutants significantly reduce host cell HR after infection with the bacterium *Pseudomonas syringae* pathovar (pv) *tomato* (*Pst* DC3000) with the effector proteins AvrRps4 or AvrRpm1 [100]. Furthermore, when infected with *Pseudomonas syringae*, Arabidopsis *atg5*, *atg10*, and *atg18a* mutant leaves do not spread cell necrosis, and the plants show obvious resistance [101].

NBR1-mediated selective autophagy is also associated with immunity against bacteria [102,103]. While *Pst*-induced autophagy promotes bacterial proliferation, NBR1-mediated selective autophagy counteracts its induction of water-soaked lesions by inhibiting the formation of the Hrp outer protein M1(HopM1) and enhances *Pst* resistance in Arabidopsis [102]. While the way in which NBR1 targets pathogens and promotes local immunity remains unknown, these findings point to a more complex function of selective autophagy proteins in cell immunity [104]. A recent study uncovers the complex ways in which pathogens interact with their hosts. The type-III effector XopL can interact with the autophagy component SH3P2 through E3 ligase activity and degrade it to promote infection, while XopL is degraded by NBR1-mediated selective autophagy [105].

## 6. Conclusions and Future Perspectives

Plant autophagy research currently focuses on fundamental research, and how to apply it to crop improvement is an issue that needs to be considered in the future. The most direct approach is to use plant growth regulators to induce autophagy. For example, exogenous spraying of melatonin can improve the heat tolerance of tomatoes; it may be that melatonin increases the expression of ATGs and the formation of autophagic vesicles at high temperatures to degrade the denatured proteins produced under heat stress [106]. Exogenous BR treatment can also enhance tomato resistance to nitrogen starvation and cold stress through brassinazole resistance 1 (BZR1)-mediated autophagy [65,107]. Ethylene (ETH) is also reported to induce ATG expression and ROS levels to promote the survival of soybeans and tomatoes under flooding and hypoxia stress [108]. In addition to the use of hormone regulation, the development of biopesticides can also directly regulate the autophagy activity of pathogens and pests to reduce disease transmission, which is also an effective means to improve crop resistance. For example, since autophagy promotes the replication of the *rice gall dwarf virus* (RGDV) in leafhoppers, spraying the autophagy inhibitor 3-MA can reduce the spread of the RGDV virus [109]. With the development of nanotechnology, it is also reported that titanium dioxide (TiO_2_) and zinc oxide (ZnO) nanoparticles are involved in the regulation of plant autophagy [110,111]. The above three methods, using external spraying reagents, provide feasibility for the application of autophagy in agricultural production. Moreover, the use of gene editing or other means of gene manipulation to regulate the *ATG* genes is also a direction that can be developed in the future.

In recent years, the role of autophagy in plant growth and development, abiotic stress, and plant–microbe interactions has been clarified in a variety of plants (Table 1). The core process of autophagy is highly conserved in eukaryotes, and the functions and regulatory networks of autophagy are specific in different species. Although many ATG proteins have been identified, there are few studies on whether these proteins have other functions. In addition, although autophagy is involved in many life processes, the way to coordinate hormonal signals to regulate plant growth and stress resistance needs further research. Furthermore, the mechanism of autophagy regulation should be continuously studied, as it not only has theoretical significance, but also has very important application value for future crop breeding.

## Figures and Tables

**Figure 1 ijms-23-11410-f001:**
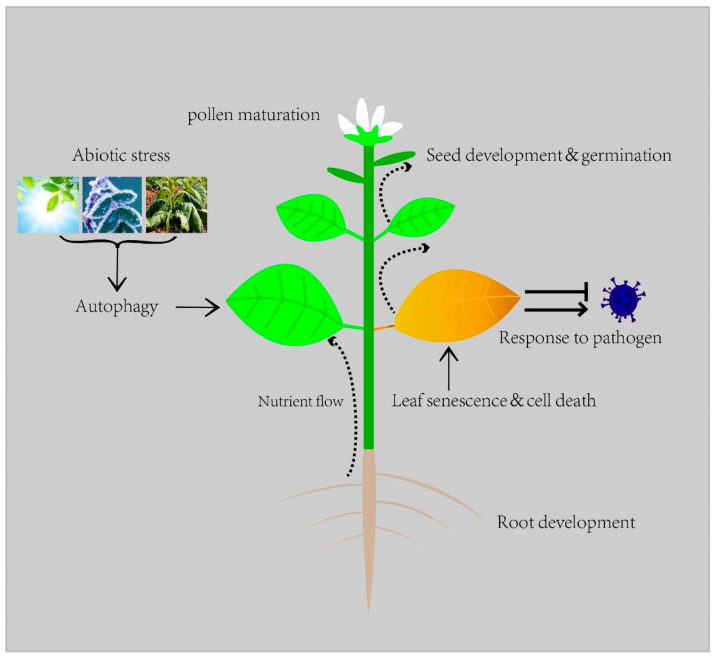
Autophagy in plant life.

**Table 1 ijms-23-11410-t001:** Plants with identified ATG genes and potential processes that require autophagy.

Species	Biological Process
*Arabidopsis thaliana*	Seed development [15], root development [23], pollen development [33,34,35], abiotic stress [63,70], and biotic stress [94,100,101,102]
*Capsicum annuum*	Abiotic stress [49]
*Hordeum vulgare*	Leaf senescence [112], nutrient remobilization [27], microspore embryogenesis [113], abiotic stress [51], and biotic stress [114]
*Manihot esculenta*	Biotic stress [115,116,117]
*Musa acuminata*	Cell death and immune response [118]
*Nicotiana tabacum*	Pollen maturation [37] and biotic stress [74,77,85].
*Oryza sativa*	Pollen maturation [36], nutrition stress [119], and leaf senescence [28]
*Triticum aestivum*	Nutrition stress [120] and biotic stress [96]
*Zea mays*	Nutrition stress [13] and lipid metabolism [121]

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
