# Peer review of "Autophagy in the Lifetime of Plants: From Seed to Seed"

_ijms, 2022, doi:10.3390/ijms231911410_

Round 1

Reviewer 1 Report

In this review, the authors summarized the role of autophagy in plant vegetative growth, reproductive growth, resistance to biotic, and abiotic stresses. Overall, this is a very nice review that comprehensively summarizes the function of plant autophagy in multiple perspectives, and discussed an emerging direction of applying it in agricultural production. The flow of the paper is well organized, and most of the references are properly cited.  For further improving this manuscript, I have the following comments/ suggestions.

1. The language of this manuscript needs to be polished. Currently, there are too many small errors/mistakes all over the place. Just using page one as an example:

1) Line 28 “recycle” should be “recycling”.

2) Line 33, “to” should be deleted.

3) Line 34 “microautophagy found” needs a “was” in middle.

4) Line 38 “proteins found” needs a “were” in middle.

5) Line 41 “material” should be “materials”.

6) Line 42 “occurs the programmed cell death” needs to add something like “during” in the middle.

7) Line 43 “developmentally” what?

These are only the obvious grammar issues on page one, not talking about the sentence structures. I leave the rest to the authors to correct, and they may need to include a native speaker for editing the language.

2. The authors need to be careful with certain statements in the manuscript. Although it is already clear in the title that this paper is talking about the plant autophagy. It will still be beneficial if they can add something like “In plants” at certain places to avoid confusion to non-expert readers. For example: Line 29, when talking about three types of autophagy, they should be aware that mega-autophagy was only discovered in plants while mammalian cells have one different type, chaperone-mediated autophagy(CMA).

Author Response

Dear editior and Reviewers,

Thank you very much.

These comments are very valuable and very helpful for revising and improving our paper, as well as the important guiding significance to our researches. We have studied comments carefully and have made correction which we hope meet with approval. The main corrections in the paper and the responds to the reviewer’s comments are as following:

Response to Reviewer 1 Comments:

Ponint 1: In this review, the authors summarized the role of autophagy in plant vegetative growth, reproductive growth, resistance to biotic, and abiotic stresses. Overall, this is a very nice review that comprehensively summarizes the function of plant autophagy in multiple perspectives, and discussed an emerging direction of applying it in agricultural production. The flow of the paper is well organized, and most of the references are properly cited.  For further improving this manuscript, I have the following comments/ suggestions.

  1. The language of this manuscript needs to be polished. Currently, there are too many small errors/mistakes all over the place. Just using page one as an example:

1) Line 28 “recycle” should be “recycling”.

2) Line 33, “to” should be deleted.

3) Line 34 “microautophagy found” needs a “was” in middle.

4) Line 38 “proteins found” needs a “were” in middle.

5) Line 41 “material” should be “materials”.

6) Line 42 “occurs the programmed cell death” needs to add something like “during” in the middle.

7) Line 43 “developmentally” what?

These are only the obvious grammar issues on page one, not talking about the sentence structures. I leave the rest to the authors to correct, and they may need to include a native speaker for editing the language.

Response 1: Thank you. We have corrected the wrong part according to your comments. In addition, we have invited a native speaker to help us revise the full text. Please see it.

Point 2. The authors need to be careful with certain statements in the manuscript. Although it is already clear in the title that this paper is talking about the plant autophagy. It will still be beneficial if they can add something like “In plants” at certain places to avoid confusion to non-expert readers. For example: Line 29, when talking about three types of autophagy, they should be aware that mega-autophagy was only discovered in plants while mammalian cells have one different type, chaperone-mediated autophagy(CMA).

Response 2: Thank you. we have corrected the ambiguity based on your comments.

Reviewer 2 Report

Paper reports interesting review of the present research on autophagy in the lifetime of plants. Authors provide an information about research progress of autophagy in plant vegetative growth, reproductive growth, and resistance to biotic and abiotic stresses. Some future perspectives are also provided.

It should be emphasized, that the paper is very well written, which shows very good organization, readability and grammar.  I have noticed only few minor mistakes, e.g.

Line 291 5.2. Autophagy and fungus – I think, instead of “fungus” there should be plural: “fungi”

Line 328 saprophytic – should be saprotrophic

Line 318 Puccinia striiformis f. sp. tritici – the abbreviation f.sp. should not be in italics

Line 320 Blumeria graminis f. sp. tritici,the abbreviation f.sp. should not be in italics

Line 416 reference 16 should be corrected following journal guidelines

Line 425 reference 19 should be corrected following journal guidelines

The section References should be carefully checked and corrected following journal guidelines. There are also few editorial mistakes. Therefore the text should be thoroughly checked and corrected before acceptance.

In my opinion, the manuscript deserve to be published in the International Journal of Molecular Science, an MDPI journal.

Author Response

Dear editior and Reviewers,

Thank you very much.

These comments are very valuable and very helpful for revising and improving our paper, as well as the important guiding significance to our researches. We have studied comments carefully and have made correction which we hope meet with approval. The main corrections in the paper and the responds to the reviewer’s comments are as following:

Response to Reviewer 2 Comments:

Point 1: Paper reports interesting review of the present research on autophagy in the lifetime of plants. Authors provide an information about research progress of autophagy in plant vegetative growth, reproductive growth, and resistance to biotic and abiotic stresses. Some future perspectives are also provided.

It should be emphasized, that the paper is very well written, which shows very good organization, readability and grammar.  I have noticed only few minor mistakes, e.g.

Line 291 5.2. Autophagy and fungus – I think, instead of “fungus” there should be plural: “fungi”

Response 1: Thank you. We have corrected it. See line 252.

Point 2: Line 328 saprophytic – should be saprotrophic

Response 2: Thank you. We have corrected it. See line 280.

Point 3: Line 318 Puccinia striiformis f. sp. tritici – the abbreviation f.sp. should not be in italics

Response 3: Thank you. We have corrected it. See line 273.

Point 4: Line 320 Blumeria graminis f. sp. tritici, – the abbreviation f.sp. should not be in italics

Response 4: Thank you. We have corrected it. See line 274.

Point 5: Line 416 reference 16 should be corrected following journal guidelines

Response 5: Thank you. We have corrected it. See line 373.

Point 6: Line 425 reference 19 should be corrected following journal guidelines

Response 6: Thank you. We have corrected it. See line 380.

Point 7: The section References should be carefully checked and corrected following journal guidelines. There are also few editorial mistakes. Therefore the text should be thoroughly checked and corrected before acceptance.

In my opinion, the manuscript deserve to be published in the International Journal of Molecular Science, an MDPI journal.

Response 7: Okay, thanks for your advice. We have checked and revised the MS based on your comments.

Round 2

Reviewer 1 Report

In this revised manuscript, the author addressed most of the concerns raised by this reviewer. Now it should be suitable for publication in IJMS.